

# Retrieval of Atmospheric Water Vapor and Temperature Profiles over Antarctica through Iterative Approach

Zhimeng Zhang[1], Shannon Brown[2], Andreas Colliander[2]

[1]California Institute of Technology, Pasadena, CA, USA

[2]Jet Propulsion Laboratory, California Institute of Technology, Pasadena, CA, USA

*Correspondence to*: Zhimeng Zhang (zhimeng@caltech.edu)

**Abstract.** Retrieving atmospheric water vapor and temperature profiles presents considerable challenges over land surfaces using microwave radiometry due to uncertainties associated with estimating background surface emissions. In response, we have devised an approach that integrates the atmospheric retrieval algorithm with the background emission algorithm, establishing an iterative loop to refine the accuracy of atmospheric profiles. Leveraging optimal estimation techniques with sounding channels spanning from Ka- to G-band obtained from ATMS, we successfully retrieved atmospheric temperature and humidity profiles across space and time. These retrieved atmospheric profiles undergo continual updates throughout each iteration, exerting influence on subsequent surface retrievals. This iterative process persists until convergence is achieved in the atmospheric retrieval. The algorithm's novelty lies in its fusion of surface retrieval with atmospheric retrieval, thereby enhancing overall accuracy. We validated the retrievals against radiosonde data. Our iterative algorithm proved to be efficient and accurate in retrieving temperature profiles with surface emissivity and in detecting melting events. Though our algorithm was able to capture the water vapor variations, the results showed that to obtain accurate absolute values of the water content an independently retrieved surface emissivity is required.

## 1    Introduction

The Greenland and Antarctica ice sheets play a crucial role in the ongoing global sea level rise due to the increasing melting of these vast ice reserves [Rignot et al. 2011]. This emphasizes the pressing need to closely monitor and comprehend the mechanisms driving the accelerated melt events taking place [Noble et al. 2020]. To effectively monitor and analyze the ice sheets, it becomes paramount to have a deep understanding of the atmospheric conditions that directly impact these regions [Le clec'h et al. 2019]. This understanding is vital for accurately predicting the future evolution of these ice masses [Le clec'h et al. 2019].

The current atmospheric retrievals encounter certain challenges, particularly concerning uncertainties in estimating the surface emission background. These uncertainties are further compounded by the unique characteristics of the polar ice sheets, which typically have a dry atmosphere, rapidly changing surface emissivity during the melting seasons, and a relatively high background emission, all of which contribute to the complexity of the problem [Miao





et al., 2001]. Traditional retrievals are hindered by the lack of independent surface emissivity and the uncertainties in
estimation of the background surface emission.
To address these challenges, we propose implementing a strategy that involves integrating the atmospheric
retrieval process with a surface emission estimation within an iterative loop. This approach aims to enhance the
accuracy and reliability of the retrieved atmospheric temperature and humidity profiles, leading to a better
understanding of the mechanisms influencing the melting of the Greenland and Antarctica ice sheets.
The Advanced Technology Microwave Sounder (ATMS) instruments on board polar orbiting satellites serve the
essential function of conducting temperature and water vapor sounding in the atmosphere [Goldberg et al. 2006, Muth
et al. 2005]. With a total of 22 channels, they are equipped to receive and measure radiation from various layers of the
atmosphere. These spectral positions include the oxygen band spanning 50–58GHz, the two distinctive water vapor
lines at 22GHz and 183 GHz, and specifically designed transparent window channels.
We implemented our iterative retrieval algorithm to analyze the ATMS data collected specifically from 17
radiosonde stations in the Antarctica region throughout the year 2016. We chose 2016 because of the extensive melt
anomaly over the Ross Ice Shelf during the 2015-2016 austral summer (e.g., Nicolas et al., 2017; Mousavi et al., 2022;
de Roda Husman et al., 2024; Hansen et al., 2024) that provides a particularly suitable conditions for testing the effect
of changing surface emissivity on the atmospheric retrievals. This analysis allowed us to examine the validity and
limitations of the iterative algorithm.

**2    Observations**
**2.1    Observation Data - ATMS data**

The Advanced Technology Microwave Sounder (ATMS) onboard polar-orbiting satellites are meant for the
atmosphere's temperature and water vapor sounding [Goldberg et al. 2006, Muth et al. 2005, Kim et al. 2014, 2020].
It has 22 channels to receive and measure radiation from different layers of the atmosphere to obtain global data on
tropospheric humidity and temperature at either quasi-vertical or quasi-horizontal polarization. Table 1 lists the
channels' radiometric characteristics, and Figure 3 shows the spectral positions of these channels with respect to the
atmospheric opacity caused by oxygen and water vapor. The cross-track scanning microwave sensors measure
microwave thermal emission from the Earth and its atmosphere in the oxygen band of 50–58GHz, the two water vapor
lines at 22 GHz and 183 GHz, and window channels (see Fig. 3), with a swath width of approximately 2600 km. The
temperature information of the atmosphere is obtained from Channels 3-15. Channels 18-22 are centered at the 183.31
GHz water vapor line but with a successively narrower bandwidth from $\pm7$GHz to $\pm1$GHz, giving humidity
information on the successively higher troposphere. The beam width of the ATMS varies across its frequency channels:
it is 1.1 degrees for channels in the 160-183 GHz range, 2.2 degrees for the 80 GHz and 50-60 GHz channels, and 5.2
degrees for the 23.8 and 31.4 GHz channels [Kim et al. 2014, 2020]. Both SNPP (Suomi National Polar-orbiting
Partnership) and the NOAA-20 (National Oceanic and Atmospheric Administration 20, also known as Joint Polar
Satellite System 1, JPSS-1) satellites orbit at an altitude of approximately 830 km. This results in instantaneous spatial
resolutions on the ground at nadir of about 16 km, 32 km, or 75 km, depending on the channel. Due to the cross-track





scanning approach, the instrument provides observations at a wide range of incidence (off-nadir) angles. To ensure
accuracy, we only use observations with an incidence angle less than 60°, as the instrument footprint increases
proportionally to $1/cos^2\theta$. The ATMS data used in this work were obtained from the Comprehensive Large Array-
data Stewardship System (CLASS) of NOAA.

Figure 1 shows the zenith opacity of a typical polar dry atmosphere for the ATMS frequency range. The opacity is
dominantly caused by water vapor and oxygen in the atmosphere. The contribution of nitrogen, orders of magnitude
smaller and without any spectral line at this frequency range, can be neglected. The total opacity is computed by
integrating the opacity per unit length from the top of the atmosphere toward the surface [Meeks and Lilley 1963].
Channels 1, 2, 16, and 17 are located at transparent window regions, where the opacity is low. The atmosphere
becomes opaque to a channel at the layer of the atmosphere where the integral of the total opacity reaches the value
of one [Petty 2004]. The Jacobians of the temperature and water vapor channels peak when the opacity becomes one.
For the window channels, the overall opacity is much less than one. These channels can see through the atmosphere
and are sensitive to Earth's surface emission; therefore, they can be used for deriving surface emissivity. However, at
oxygen (58 GHz) and water vapor (183 GHz) absorption bands, the observed brightness temperature is not very
sensitive to the surface emission. The surface emissivity at these frequencies will be derived using the adjacent
channels that have relatively low opacity: CH3 at 50.3 GHz will be used to derive the emissivity in the range of 50.3-
57.29 GHz (CH3 to CH15) at QH polarization; CH18 at 183.31±7 GHz for the emissivity near 183.31 GHz (CH18
to CH22) at QH polarization. In all, we derive surface emissivity for the window channels 1 (23.8 GHz), 2 (31.4 GHz),
3 (50.3 GHz), 16 (88.2 GHz), 17 (165.5 GHz), and 18 (183.31±7GHz) and assume the emissivity variation is linear
between the frequencies.

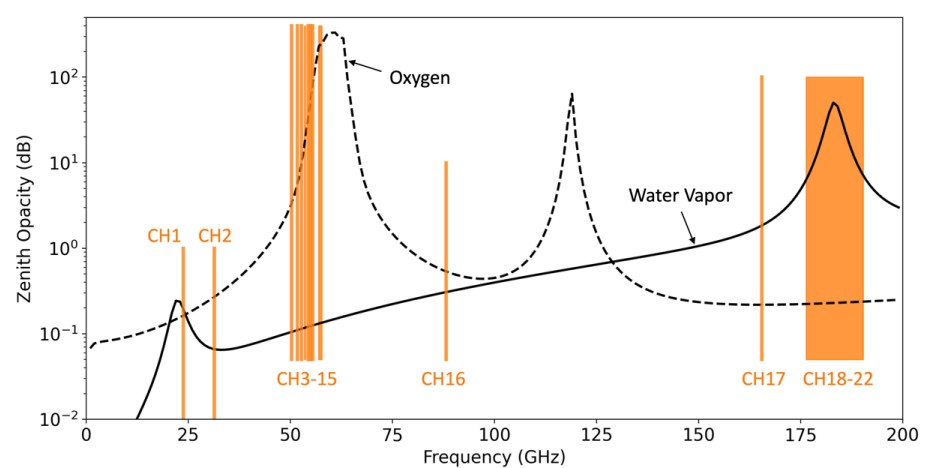


**Figure 1.** Spectra positions of 22 ATMS channels, on top of zenith opacity due to oxygen and water vapor.





**Table.1**

| Channel Number | Central frequency [GHz] | $NE\Delta T$ [K] | Polarization |
|---|---|---|---|
| 1 | 23.8 | 0.7 | QV |
| 2 | 31.4 | 0.8 | QV |
| 3 | 50.3 | 0.9 | QH |
| 4 | 51.76 | 0.7 | QH |
| 5 | 52.8 | 0.7 | QH |
| 6 | 53.596 ± 0.115 | 0.7 | QH |
| 7 | 54.4 | 0.7 | QH |
| 8 | 54.94 | 0.7 | QH |
| 9 | 55.5 | 0.7 | QH |
| 10 | 57.290344 | 0.75 | QH |
| 11 | 57.290344 ± 0.217 | 1.2 | QH |
| 12 | 57.290344 ± 0.3222 ± 0.048 | 1.2 | QH |
| 13 | 57.290344 ± 0.3222 ± 0.022 | 1.5 | QH |
| 14 | 57.290344 ± 0.3222 ± 0.010 | 2.4 | QH |
| 15 | 57.290344 ± 0.3222 ± 0.0045 | 3.6 | QH |
| 16 | 88.2 | 0.5 | QV |
| 17 | 165.5 | 0.6 | QH |
| 18 | 183.31 ± 7.0 | 0.8 | QH |
| 19 | 183.31 ± 4.5 | 0.8 | QH |
| 20 | 183.31 ± 3.0 | 0.8 | QH |
| 21 | 183.31 ± 1.8 | 0.8 | QH |
| 22 | 183.31 ± 1.0 | 0.9 | QH |

Figure 2 shows the temperature (third column) and water vapor (fourth column) Jacobians at all 22 ATMS channels for June (austral winter; upper row) and December (austral summer; lower row) atmospheres, representative of Antarctica atmosphere. The Jacobians are calculated by increasing the water vapor and temperature at each vertical layer to the orange dashed line. Therefore, a positive value implies that adding water vapor or increasing temperature will increase the radiance observed by the instrument and vice versa. The Jacobian shows the sensitive altitudes of each channel. Near the oxygen absorption band, CH3 to CH15 are sensitive to successively higher altitudes, up to ~60 km above the sea level. Channels 18–22 are located near the water vapor line at 183.31 GHz. The atmospheric opacity for these channels differs significantly, making these channels sensitive to different layers of the atmosphere. The bandwidth of these channels decreases with the channel number, with Channel 22 being the narrowest and Channel 18 being the broadest. Water vapor Jacobians for Channels 18–22 show that the channels are sensitive to different layers of the atmosphere with a sensitivity of about a few kilometers. For the atmosphere above ~10 km, the water



vapor content drops so low that common perturbations cannot be detected by ATMS. The Jacobians shown are from
the nadir view (0° incidence angle). Jacobians corresponding to off-nadir angles peak at slightly higher altitudes
because of the longer path length the radiation must travel.

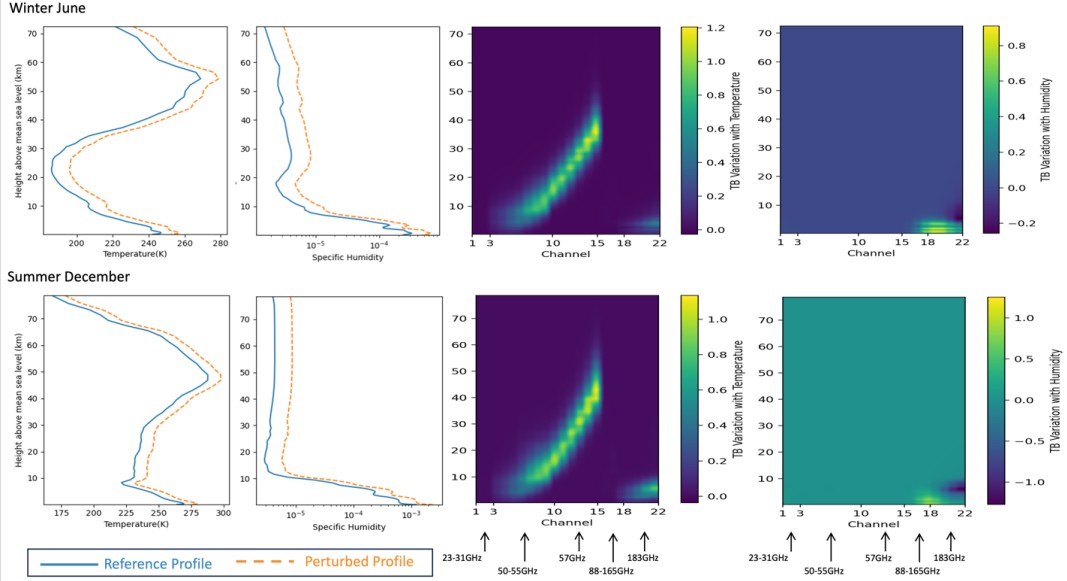


**Figure 2.** Typical Antarctica atmospheres for winter (upper row) and summer (lower row) weathers at WAIS DIVIDE
(79.5°S, 112°W). The plots show the temperature profile (left column) and the specific humidity profile (second
column), the temperature Jacobian (third column), and the water vapor Jacobian (forth column) for all 22 ATMS
Channels. The Jacobians are based on nadir view (0° incidence angle).

**2.2    The Prior – MERRA-2**

With ATMS capturing microwave thermal emission, our approach involved using Optimal Estimation (OE) to
transform these observations into the desired variable of interest. OE, as outlined by Maahn et al. (2020), acts as a
widely accepted physical retrieval method that takes into account various factors such as measurements, prior
information, and associated uncertainties. By leveraging Bayes' theorem, OE aims to derive the most efficient solution
for determining the atmospheric state, particularly when retrieving temperature and humidity profiles from Microwave
Radiometer (MWR) data. The role of the prior information in this process is crucial. We chose to use the Modern-Era
Retrospective analysis for Research and Applications, version 2 (MERRA-2) [17] atmospheric reanalysis as the prior
for the pressure, temperature, water vapor, and liquid water profiles (e.g., Rosenkranz. 2001). This dataset, known as
"tavg3_3d_asm_Nv," provides a comprehensive overview of the atmosphere with its three-hour averaged structure
comprising 72 vertical layers and a spatial resolution of 0.625 degrees in longitude and 0.5 degrees in latitude. To
account for surface characteristics such as skin temperature, temperature/wind speed at two meters, and surface



pressure, we integrated MERRA-2 single-level diagnostics data labeled as "tavg1_2d_slv_Nx". By merging the three-
dimensional assimilated meteorological data with the single-level diagnostics data, our analysis extends across 73
vertical layers starting from two meters above the surface and extending beyond 70 kilometers into the atmosphere.

**2.3     Radiosonde data**

In the Antarctica region, a network of radiosonde stations is strategically positioned to capture in situ measurements
of atmospheric profiles. These stations contribute to the Integrated Global Radiosonde Archive (IGRA), collected
once per day, albeit at a relatively low time resolution [Durre et al. 2006]. However, efforts such as the Atmospheric
Radiation Measurement (ARM) West Antarctic Radiation Experiment (AWARE) have bolstered observational
capabilities by providing high temporal resolution radiosonde data at key locations such as the West Antarctica Ice
Sheet (WAIS) and McMurdo [Lubin et al. 2017]. During 2016, AWARE collected atmospheric profiles every minute,
offering unprecedented insights into the dynamic atmospheric processes in this critical region. Observing stations in
Antarctica that measure ground-based meteorological data are located mostly around lower elevations and coastal
areas. We only found two stations on the high plateau (above 2500 m), including South Pole Station (90° S) [Xu et al.
2019] and WAIS DIVIDE (79° S) [Lubin et al. 2017], that have measurements during 2016. The radiosonde
measurements reveal detailed features in temperature and humidity profiles at high altitude resolution ranging from
20 m to a few hundred meters.

In our study, an evaluation of our iterative retrieval algorithm was conducted across the network of nine radiosonde
stations in the Antarctica region throughout 2016. This testing allowed us to assess the algorithm's validation and
limitations under diverse atmospheric conditions, ice sheet melting conditions, and geographical settings. We
compared the retrievals with the temperature and humidity profiles gathered from the radiosondes. This comparative
analysis served as a benchmark for evaluating our iterative retrieval approach. Through this validation process, we
gained insights into the algorithm's efficacy in capturing the atmospheric properties and ice sheet conditions across
the Antarctica region. Furthermore, the validation results guided refining and optimizing our retrieval methodology
for enhanced accuracy and utility in atmospheric studies and ice sheet melting events assessments.

**Table.2**

| IGRA Daily | | |
|---|---|---|
| AMUNDSEN-SCOTT | 90.0° S | 0° |
| NOVOLAZAREVSKAJA | 70.7678° S | 11.8317° E |
| SYOWA | 69.0053° S | 39.5811° E |
| DAVIS | 68.5744° S | 77.9672° E |



| | | |
|---|---|---|
| MIRNYJ | 66.5519º S | 93.0147º E |
| CASEY | 66.2825º S | 110.5231º E |
| MARIO ZUCHELLI STATION | 74.6958º S | 164.0922º E |
| MCMURDO | 77.85º S | 166.6667º E |
| **ARM per minute** | | |
| McMurdo Station | 77.85º S | 166.66º E |
| WAIS Divide | 79.468º S | 112.086º W |


## 3    Iterative Retrieval Algorithm

### 3.1 Iterative Process

We developed a channel-selective iterative approach to retrieve surface emissivity along with the atmospheric temperature and humidity profiles, as illustrated in Figure 3. The procedure begins with an initial estimation of surface emissivity, which is then used to derive temperature and humidity profiles. Next, these profiles are utilized to refine the surface emissivity in a continuous loop until the update in emissivity reaches a margin of 0.01. The final discrepancy between observed and modeled brightness temperatures (the residual) is compared against the noise equivalent delta temperature to assess the accuracy. The noise equivalent delta temperature (NEΔT) indicates the level of instrument noise, which can be defined using the ideal noise equation for a total power radiometer such as ATMS [Ulaby et al. 1981]:

$$\Delta T = T_{sys} \left[ \frac{1}{B\tau} + \left( \frac{\Delta G}{G} \right)^2 \right]^{\frac{1}{2}} \quad .... [1]$$

where $T_{sys}$ is the system noise temperature (including atmosphere contribution), $B$ is the bandwidth, $\tau$ is the integration time and $\Delta G/G$ represents instrument gain fluctuations. Any residual falling within 1.5 times the noise equivalent temperature is deemed successful.

The outcome of the routine comprises the profiles of temperature and humidity as well as the surface emissivity for all frequency channels (23GHz to 183GHz). Choosing an initial emissivity value of $\varepsilon_0$ as 0.8 across all frequencies aligns with the average AMSU-A surface emissivity data in the Antarctic region used in the past [Spencer and William 1999]. In Figure 4, we show the histogram of the AMSU-A recorded surface emissivity at 23GHz, 31GHz, and 50GHz (Ferraro et al. 2016), with a median value near 0.8. The AMSU-A emissivities are directly calculated from the satellite observations in clear-sky conditions. Typically, the iterative process converges within six iterations, enabling efficient retrieval of the atmospheric temperature and humidity profiles along with the surface emissivity.





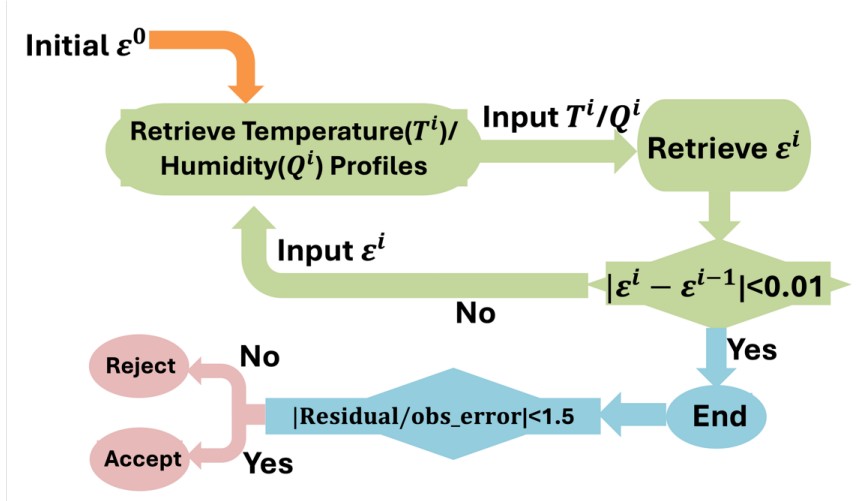

187

**Figure 3.** Flow diagram of the iterative retrieval process. The process starts with an initial guess of surface emissivity to retrieve temperature and humidity profiles, which are then utilized to adjust the surface emissivity for subsequent iterations. Residuals are determined as the disparity between observed and modeled brightness temperature and assessed against the observation noise equivalent temperature. Any outcomes with residuals surpassing 1.5 times the noise are regarded as unsuccessful and discarded. The final outcomes are the temperature and humidity profiles alongside the surface emissivity.



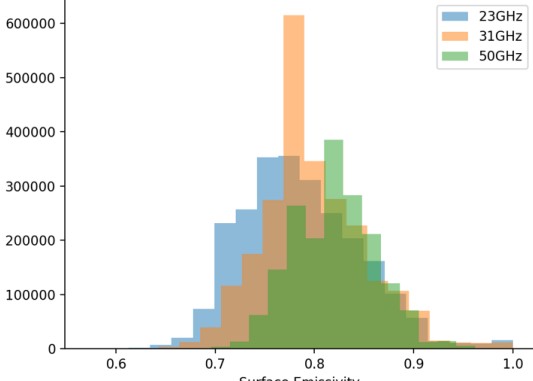


**Figure 4.** In the iterative retrieval process, the starting value of $\varepsilon0$ is set at 0.8 for all frequencies based on AMSU-A surface emissivity data in the Antarctic region. The histogram depicting AMSU-A surface emissivity records at 23GHz, 31GHz, and 50GHz shows a median value near 0.8.




**3.2    Forward Model - Radiative Transfer Model**

RTTOV (Radiative Transfer for TOVS[1]) is a radiative transfer code widely used in atmospheric remote sensing
research [Saunders et al. 2018]. The model incorporates detailed representations of atmospheric properties, including
temperature, humidity, pressure, and optionally, trace gases, aerosols, and hydrometeors, together with surface
parameters and viewing geometry, allowing for accurate simulations of radiative transfer processes in diverse
atmospheric conditions. RTTOV is a fast radiative transfer model for passive visible, infrared, and microwave
downward-viewing satellite radiometers, spectrometers, and interferometers. RTTOV computes the top-of-
atmosphere radiances in each of the sensor channels being simulated. We incorporate a Python interface written for
RTTOV v11.3. The required geophysical inputs of the model are humidity and temperature profiles, the surface skin
temperature, and the surface emissivity. The model was not only used to calculate radiances but also to calculate the
associated Jacobians:
$$K_j(\theta) = \frac{\partial T_B(\theta)}{\partial x_j} \quad \ldots[2a]$$
where j is the vertical grid index and $x_j$ can be the water vapor mass mixing ratio (Q) in fractional units or temperature
(T) in Kelvin
$$x_j = \frac{Q_j^{H_2O}}{Q_j^{Ref}} \text{ or } x_j = T \; \ldots[2b]$$
The $Q_j^{Ref}$ are identical to the profile for which the Jacobian is calculated. This type of Jacobian shows the sensitivity
of $T_B$ to relative changes in the humidity/temperature at each vertical grid point. The profile is assumed to be linear
between the vertical grid points. The grid used is equidistant in the logarithm of the pressure; hence, it is approximately
equidistant in altitude.

**3.3 Retrieved Profiles**

We utilized our iterative retrieval method, as described in Section 3.1, on the ATMS data collected over the Antarctica
region during 2016. Figure 5 displays an instance of the evolution of the retrieved profiles through multiple iterations,
including the temperature profiles (top left), specific humidity profiles (top right), and frequency-dependent surface
emissivity (bottom left) acquired after six iterations. The bottom right panel shows the residual divided by the
observation error. As discussed in Section 3.1, if the residuals for all frequencies/channels are under 1.5 times the
uncertainty, the model is considered to fit well. Following six iterations, we observe convergence in the iterative
process.

---

[1] **TOVS:** Television InfraRed Observation Satellite (TIROS) Operational Vertical Sounder, a suite of three
instruments that measure upwelling radiation from the atmosphere from which surface properties, clouds, and the
vertical structure of the atmosphere can be determined.



The graphs show temperature profiles up to 20 km above sea level and specific humidity profiles up to 10 km above
sea level, where radiosonde data is accessible. In the upper two panels, the purple curves display the radiosonde *in*
*situ* measurements, used to validate the iterative retrieval algorithm. The radiosonde measurements reveal more
detailed features in temperature and humidity profiles at higher altitude resolutions ranging from 20 m to a few
hundred meters, which the microwave radiometer based algorithms are unable to retrieve. Despite these fine features,
the temperature profile obtained through iterations closely matches the radiosonde measurements. The specific
humidity below 6 km also shows a good agreement with the measurements. However, our retrieval method is not as
sensitive to specific humidity above 6 km, aligning with the contribution function in Fig. 2 and the lower humidity
levels observed above 6 km.

The brightness temperature coming from both atmospheric and surface emission can be calculated using Equation 3:
$$L(\nu) = \int_{\tau_s}^{1} B(\nu, T) d\tau + [1 - \varepsilon_s(\nu)] \cdot \tau_s^2(\nu) \cdot \int_{\tau_s}^{1} \frac{B(\nu,T)}{\tau^2} d\tau + \varepsilon_s(\nu) \cdot \tau_s(\nu) \cdot B(\nu, T_s) , \quad \dots[3]$$


where $L$ is the radiance at frequency $\nu$; $B(\nu, T)$ is Planck's law; $\tau$ is the transmittance from the top of the atmosphere,
and $\tau_s$ is the transmittance from the top of the atmosphere to the surface, both of which depend mostly on oxygen and
water vapor absorption; $\varepsilon_s$ is the surface emissivity, while $1 - \varepsilon_s$ is the respective surface reflectivity, and $T_s$ is the
surface skin temperature. The three terms on the right-hand side correspond to the upwelling atmospheric emission,
the downwelling atmospheric emission reflected by the surface, and the surface emission respectively.

When the atmospheric profiles are known, the observed brightness temperature from the top of the atmosphere can be
represented as a linear function of the surface emissivity. Therefore, by utilizing the atmospheric profiles measured
by the radiosondes and the brightness temperature observed by ATMS, we can solve the surface emissivity at the
ATMS observation frequencies. This is considered as the reference surface emissivity here (the black dotted line in
the lower left panel in Figure 5). Upon comparison with our iteratively retrieved surface emissivity (red curve), we
discovered that our retrieved results are highly consistent with the reference values. However, at 183 GHz (water
vapor absorption band), the surface emission becomes closely intertwined with the emission from near-surface water
vapor, making it challenging to accurately determine the value using our iterative algorithm. Consequently, our
iterative algorithm may not always provide accurate results at 183 GHz (see Section 4.1). By using the reference
surface emissivity, the retrieved profiles displayed in the black dotted lines in the upper panels of Fig. 5 represent the
best atmosphere profiles achievable with the ATMS 22 channels observations (optimal profile). The comparison
between the optimal profile and the iterative retrieved profile in the upper panels indicated a high level of consistency.

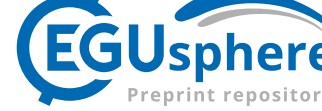

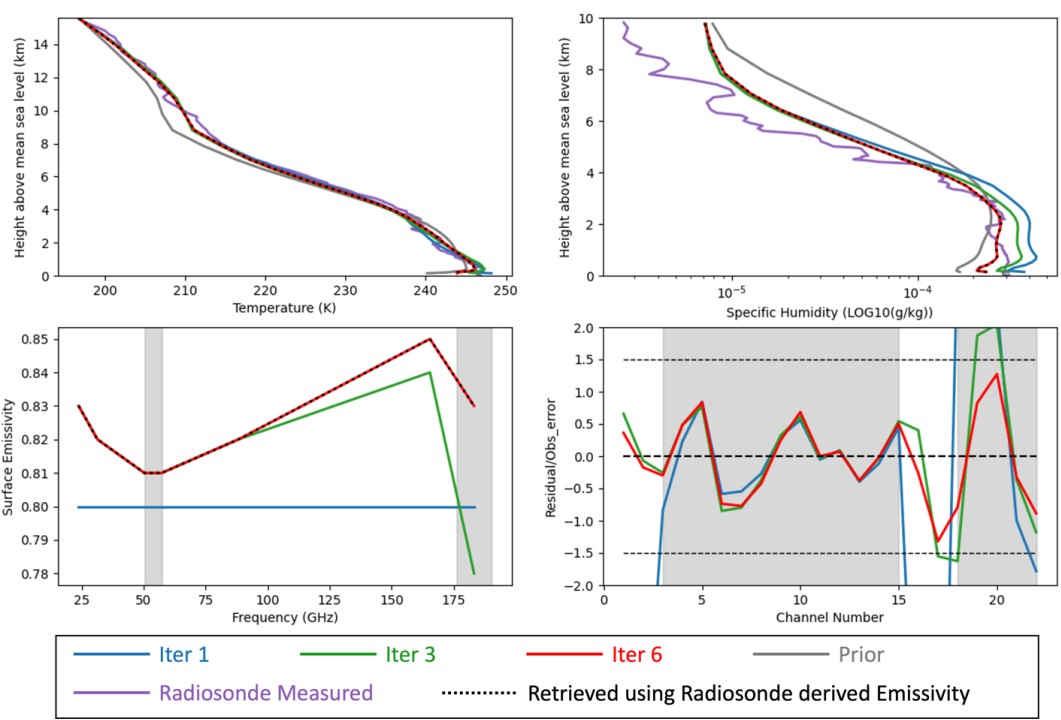

**Figure 5.** Retrieved temperature profiles (top left), specific humidity profiles (top right), and frequency-dependent surface emissivity (lower left) over six iterations (for clarity, only iterations 1, 3, and 6 are displayed). The lower right panel shows the residual divided by the observation error (the model's fit is considered good when the absolute residuals at all frequencies are less than 1.5 times the uncertainty, indicated by the dashed lines). The iteration process converges within six iterations. The gray curve shows the prior for the temperature and humidity profiles. In the bottom two figures, the oxygen and water vapor absorption frequency bands/channels are shaded in gray. In the upper two panels, the purple curve shows the radiosonde *in situ* measurements. The black dotted line shows the retrieved profiles using the radiosonde-derived emissivity.

## 4    Validation and Discussion

### 4.1 McMurdo Station

The findings over the radiosonde stations highlighted the changes in surface and atmospheric conditions over the year and their connections. We compared the retrieved profiles with the radiosonde measurements to demonstrate the effectiveness and constraints of our iterative retrieval method.

The McMurdo Radiosonde Station is located on Ross Island at coordinates 77.85°S, 166.66°E. In 2016, radiosonde observations were obtained with high time resolution at this station. The Figure 6 histogram displays the variance



between the atmospheric profiles derived by our iterative algorithm and the optimal profiles (as explained in Section
3.1). The left panel depicts the temperature difference below 70 km height, while the right panel shows the specific
humidity difference below 10 km height, spanning the entire year of 2016. Each panel shows the median value and
the 68% confidence intervals (one standard deviation, or sigma). The accuracy of the temperature profile retrieval was
high, with 68% falling within ±0.5K of the reference value. However, in the right panel, while most humidity
variations were detectable (to be further discussed in the next paragraph), the absolute value was not precise compared
to the measurement, with only 68% showing a difference within 30% of the actual profile. The humidity was strongly
linked to the surface emissivity at 183 GHz. Fig. 7 shows the histogram displaying the difference between the surface
emissivity retrieved iteratively and the reference value. The surface emissivity retrieved iteratively is highly accurate
within a range of ±0.01 for clear-sky transparent window channels like 23.8 GHz, 31.4 GHz, and 88.2 GHz. In the
oxygen absorption band between 50 and 58 GHz, most of the retrieved surface emissivity values fall within ±0.02 of
the reference value, aligned with the fact that retrieved temperatures have a one-sigma difference within ±0.5K from
the actual value. The channels with frequencies exceeding 165 GHz are impacted by the water vapor absorption band;
specifically, at 165GHz, only 68% of the emissivity values differ from the actual value within 0.03, and at 183GHz,
only 68% of the emissivity values differ from the reference value within 0.1. These discrepancies are too significant
to consider the retrieved humidity profile satisfactorily accurate. It appears that an independent estimation of the ice
sheet surface emissivity would be needed to separate the effects of ice sheet emissions and atmospheric humidity.

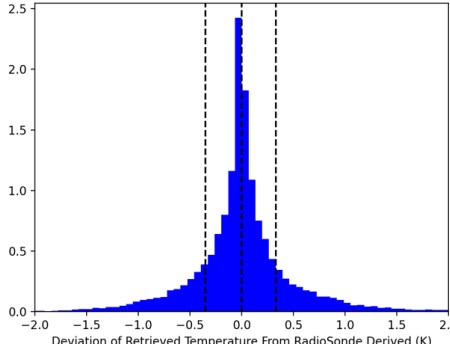
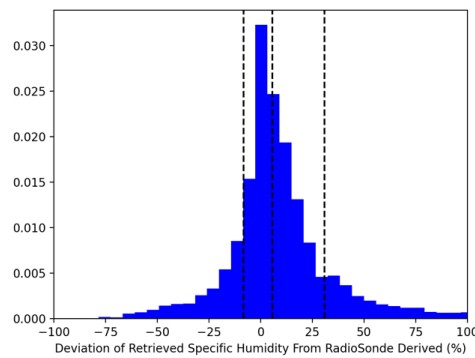


**Figure 6.** Histogram of the difference between the atmospheric profiles retrieved by our iterative algorithm and the
optimal profiles based on true surface emissivity from Radiosonde measurements. The left panel showed temperature
below 70 km and the right panel showed specific humidity below 10 km for the entire year of 2016. Dashed black
lines represented the median value and the 68% confidence intervals (one standard deviation) in each panel.





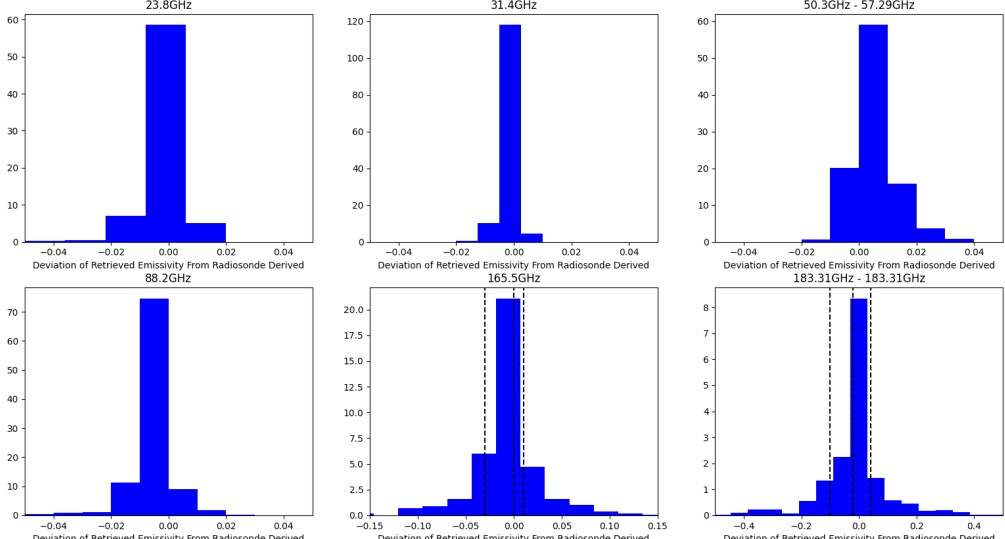

**Figure 7.** Histogram of the surface emissivity retrieved in our study compared to the reference value. The dashed black lines represented the median value and the 68% confidence intervals (one standard deviation) in the lower middle and lower right panels.

The changes in surface emissivity over the year can provide information about melting events on the ice sheet. Fig. 8a presents surface emissivity from November 2015 to the end of 2016 at three transparent window frequencies and near the oxygen absorption band. The blue curve represents values obtained from our iterative algorithm, while the red dashed line shows the reference values. These values are in good agreement with each other. Two melting events were observed around mid-December and mid-January, as indicated by the black arrows. The melting event in mid-January was more significant. Following the melting event, the emissivity decreased due to an ice crust formed within the surface snow during the refreezing. The formation of an ice crust within the surface snow increases scattering and depresses brightness temperature, resulting in reduced effective emissivity. The increase in surface emissivity during the melting event was more pronounced in the high-frequency channels (corresponding to emissivity from a more shallow surface layer), suggesting minor melting near the surface. Fig. 8b displays temperature and specific humidity perturbations from December 2015 to March 2016, encompassing the time frame of the melting events. These perturbations show deviations of the profiles from the average values during the specified period. The top row shows results from our iterative algorithm, while the bottom row shows profiles measured using radiosonde. Despite the higher altitude resolution in radiosonde measurements, our iterative algorithm captures the temperature and humidity variations effectively. The two melting events in mid-December and mid-January corresponded with increases in near-surface temperature and humidity, as indicated by the black arrows. The temperature and humidity changes during mid-January were more significant, corresponding to a more substantial melting. Temperature increased by over 10





K during the mid-January melting event, while a temperature rise of approximately 5 K was observed during the mid-
December melting event.

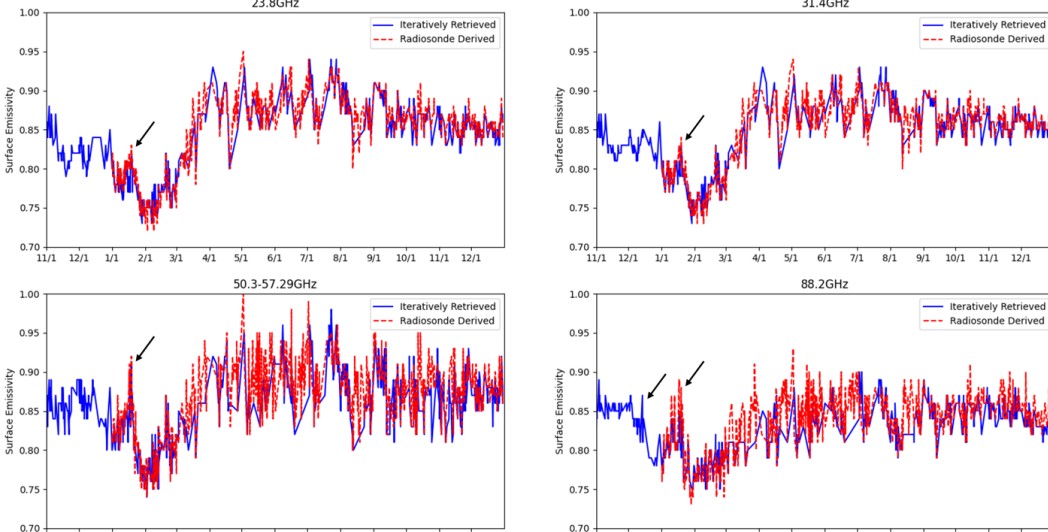


**Figure 8a.** McMurdo Radiosonde Station (77.85ºS, 166.66ºE). Surface emissivity from November 2015 to the end of
2016 at three window frequencies (23.8GHz, 31.4GHz and 88.2GHz) and the oxygen absorption band (50GHz –
58GHz). The blue curve represents values from our iterative algorithm and the red dashed line shows values derived
from the radiosonde measurements. Two melting events occurred in mid-December and mid-January, pointed out by
the black arrows. The X-axis shows the month of the year, from 2015/11/1 to 2016/12/31.




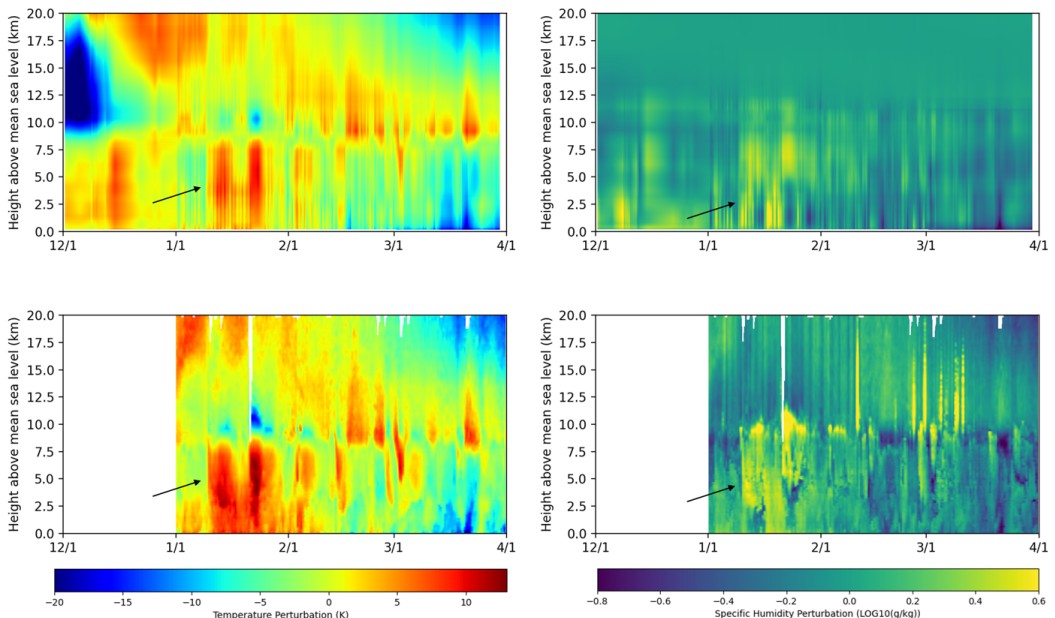

**Figure 8b.** McMurdo Radiosonde Station (77.85ºS, 166.66ºE). Temperature and specific humidity perturbations during December 2015 to March 2016. The deviations from average values are displayed for that period. The top row presents results from our iterative algorithm, while the bottom row shows data from radiosonde measurements. Melting events in mid-December and mid-January correspond to the temperature and humidity increases around the same time. The X-axis shows the month of the year, from 2015/12/1 to 2016/3/31.

**4.2 More locations near Ross Ice Shelf during the Melting Event**

**WAIS Divide Station**

The WAIS Divide Station is located at coordinates 79.468ºS, 112.086ºW, with an altitude of approximately 1.8 km above sea level. Detailed radiosonde measurements were taken between January 1st and January 15th, 2016. Fig. 9a presents the retrieved surface emissivity, and Fig. 9b depicts the temperature and specific humidity fluctuations throughout January. By utilizing the iterative retrieval method, we were able to accurately track the rise in temperature and humidity from January 10th to January 15th, aligning with the melting of the Ross Ice shelf as recorded at McMurdo Station in mid-January. In contrast to the surface emissivity changes observed at McMurdo Station, no substantial variations in surface emissivity were noted at the WAIS Divide Station, indicating stable ice crust structures and limited snowfall during the summer months. Even amidst the increase in surface temperatures, the surface temperature is still too low for melting to occur.



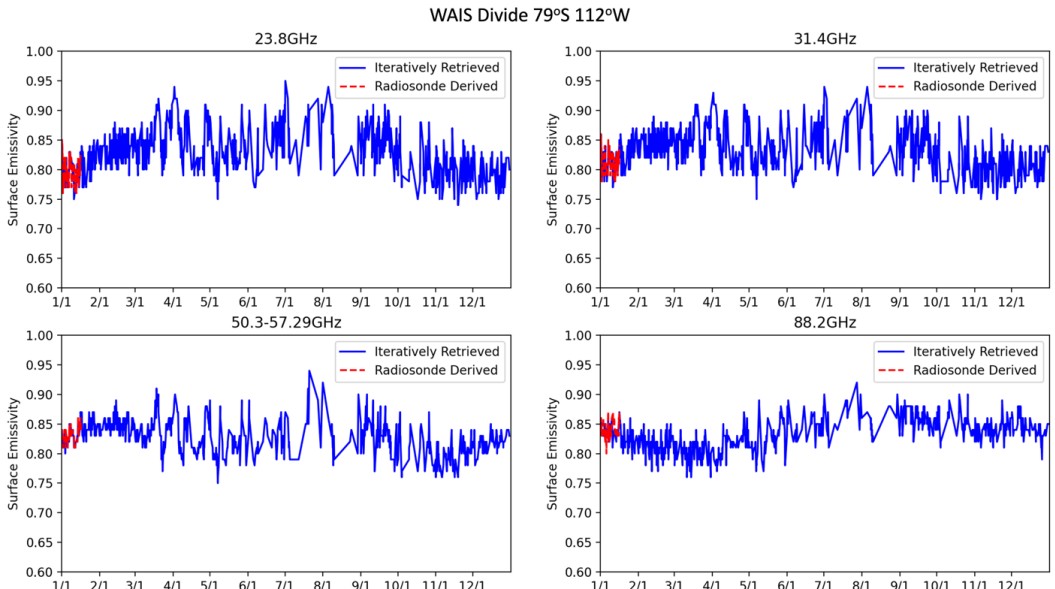

357

**Figure 9a.** WAIS Divide Radiosonde Station (79.468°S, 112.086°W). Surface emissivity from January 2016 to the
end of 2016 at three window frequencies (23.8GHz, 31.4GHz and 88.2GHz) and the oxygen absorption band (50GHz
– 58GHz). The blue curve represents values from our iterative algorithm and the red dashed line shows values derived
from the radiosonde measurements. The X-axis shows the month of the year, from 2016/1/1 to 2016/12/31.

362

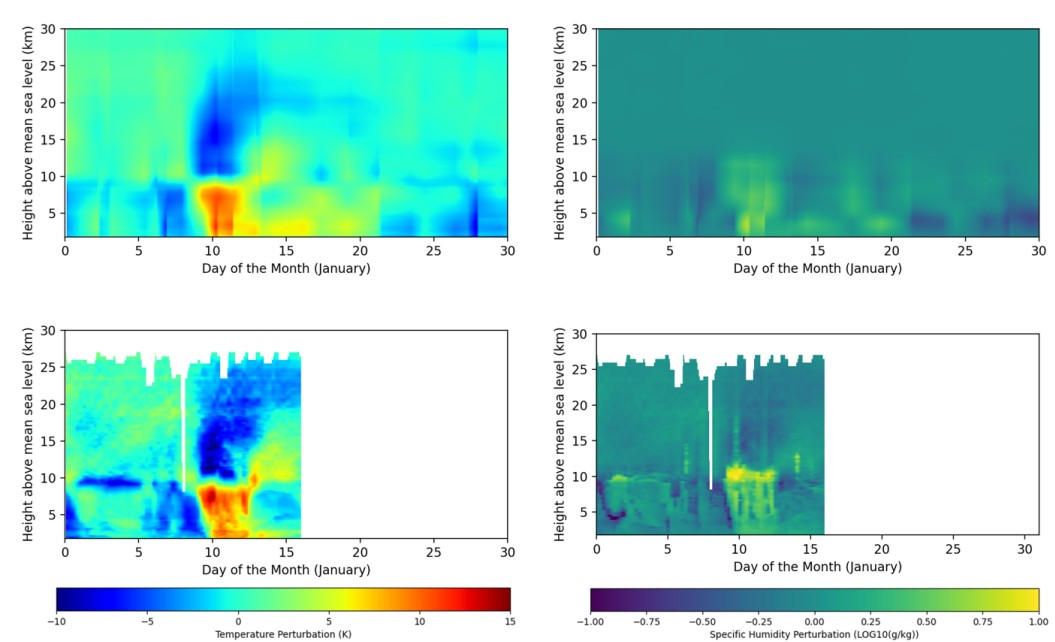

363

**Figure.9b** WAIS Divide Radiosonde Station (79.468ºS, 112.086ºW). Temperature and specific humidity perturbations during January 2016. The deviations from average values are displayed for that period. The top row presents results from our iterative algorithm, while the bottom row shows data from radiosonde measurements. The X-axis shows the month of the year, from 2016/1/1 to 2016/1/31.

368

**Ross Ice Shelf**

370

In January 2016, a significant surface melting event was documented that impacted a significant section of the Ross Ice Shelf (Nicolas et al., 2017; Mousavi et al., 2022; de Roda Husman et al., 2024; Hansen et al., 2024). The melting is associated with the strong and continuous transport of warm marine air to the region, likely influenced by the concurrent powerful El Nino event. The broad atmospheric fluctuations during and following the melting incident are found to be established by the El Nino event and mitigated by the positive Southern Annular Mode. These atmospheric changes are identified by our iterative method, as illustrated in Fig. 8b and Fig. 9b. We present findings at the coordinates 83ºS, 153ºW at the center of the melting incident. As indicated in Figure. 10a, there is a sharp rise in surface emissivity from 23.8 to 58 GHz, indicating surface melting over ice sheets and the presence of liquid water in the snowpack. Our iterative analysis of temperature and humidity perturbations in Fig. 10b captures the significant and vertically extensive intrusion of marine air from 10-13 January.

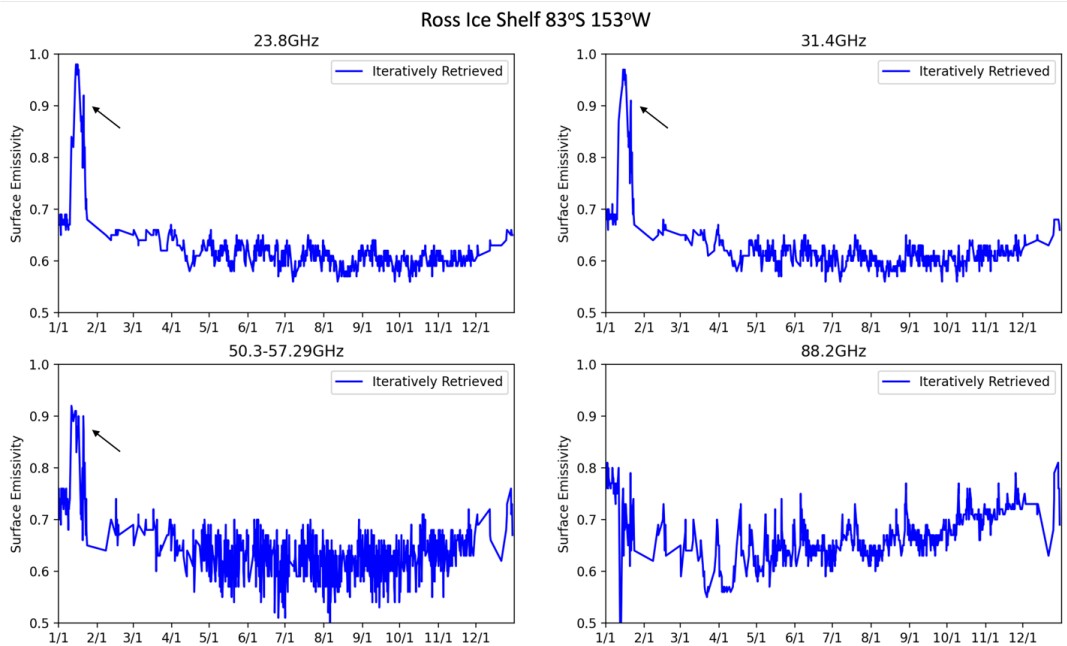

**Figure 10a.** Ross Ice Shelf (83°S, 153°W). Surface emissivity from January 2016 to the end of 2016 at three window frequencies (23.8, 31.4 and 88.2GHz) and the oxygen absorption band (50 – 58 GHz). The blue curve represents values from our iterative algorithm. The X-axis shows the month of the year, from 2016/1/1 to 2016/12/31. The melting events occurred in mid-January, pointed out by the black arrows.

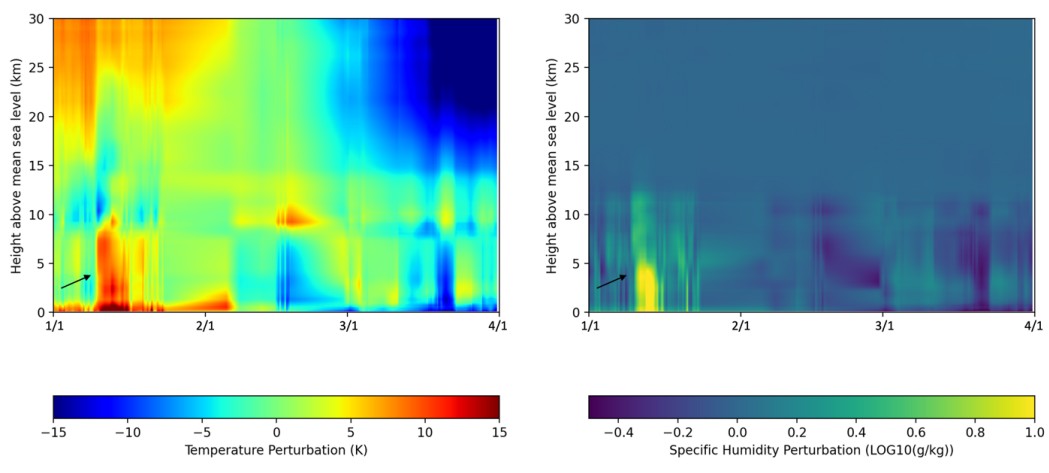

**Figure 10b.** Ross Ice Shelf (83°S, 153°W). Temperature and specific humidity perturbations during January 2016 to March 2016. The deviations from average values are displayed for that period. The results are retrieved using our iterative algorithm. The X-axis shows the month of the year, from 2016/1/1 to 2016/3/31.



390

**4.3 More Ice Shelves**

In the Antarctica region, ice shelves cover approximately 1.5 million square kilometers and include major formations such as the Ross Ice Shelf (see Section 3.3 for more details), the Ronne-Filchner Ice Shelf, and the Larsen Ice Shelf. Ice shelf melt can lead to an accelerated ice discharge from the land into the ocean, significantly contributing to the global sea level rise. Melting events over these ice shelves are, therefore, critical to understanding and predicting future sea level changes.

**West Coast Ice Shelves**

The west coast of Antarctica is home to several ice shelves, including the Larsen C Ice Shelf, Wilkins Ice Shelf, and George VI Ice Shelf, all situated on the Antarctic Peninsula. These ice shelves exhibit similar annual variations in surface properties. Notably, multiple increases in surface emissivity are observed throughout the year (Figure 11, lower left panel). These changes in emissivity correlate with increases in atmosphere temperature and humidity (Figure 11, upper panels). These increases in emissivity have different causes. The lower middle panel of Figure 11 displays the skin temperature data from ERA5 [Hersbach et al., 2023], with the black dashed line representing the melting point at 273K. During the summer months (January to March and November to December), increased emissivity corresponds to elevated skin temperatures reaching the melting threshold. An additional melting event detected in early May is attributed to concurrent rises in both near-surface atmospheric temperature (Figure 11, upper left) and skin temperature (Figure 11, lower middle). Conversely, the increased emissivity observed in late May is due to increased rain (Figure 11, lower right). These findings underscore the complex interplay of climatic factors driving the surface emissivity change on the Antarctic Peninsula's ice shelves, highlighting the sensitivity of these regions to both temperature and precipitation changes.

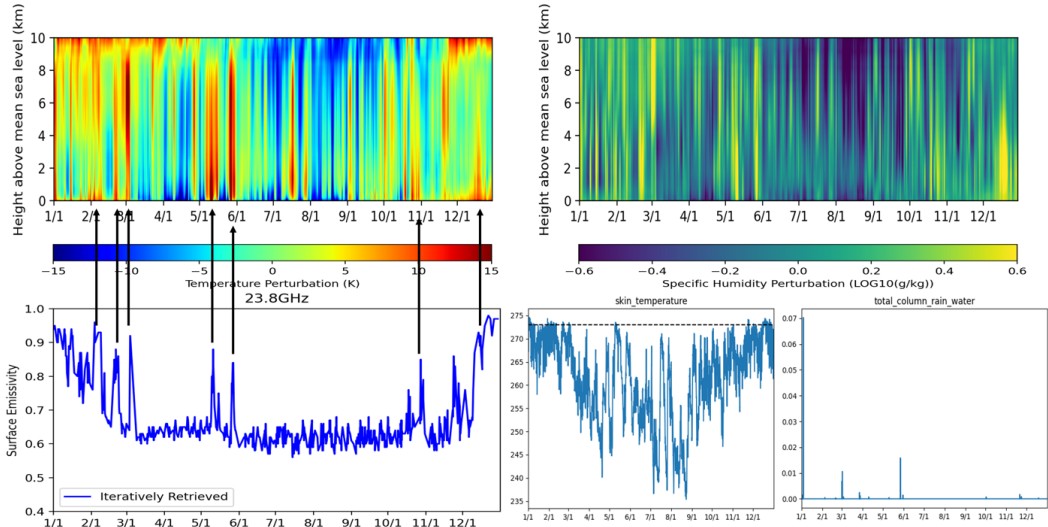

**Figure.11** Larsen C Ice Shelf (67.5°S, 62.5°W). Temperature (upper left) and specific humidity (upper right) perturbations throughout the year of 2016. The deviations from average values are displayed. Surface emissivity variation at 23.8GHz (lower left) shows a few significant increases during summer months and May. The results are retrieved using our iterative algorithm. Skin temperature (lower middle) and precipitation of rain (lower right) from ERA5 are presented. The X-axis shows the month of the year, from 2016/1/1 to 2016/12/31.

**East Coast Ice Shelves**

East coast ice shelves, such as the Amery Ice Shelf, West Ice Shelf, and Shackleton Ice Shelf, exhibit similar yearly variations in surface conditions. Unlike the complex interactions observed around the west coast, potential melting events on the east coast—indicated by increases in surface emissivity as shown in Figure 12 (lower left panel)—are detected exclusively during the summer months of January and December. According to ERA5 data (Figure 12, lower right panel), the east coast experiences extremely dry conditions with nearly zero precipitation of rain. These melting events are attributed solely to the increase in near-surface atmospheric temperature and skin temperature, which rise above the melting point during the summer.



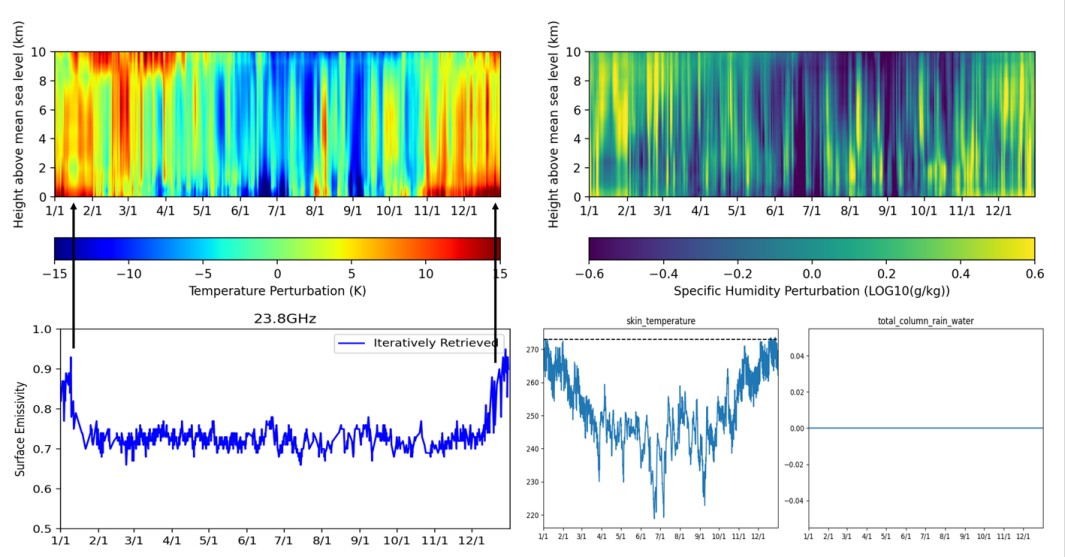

**Figure.12** Amery Ice Shelf (70ºS, 71ºE). Temperature (upper left) and specific humidity (upper right) perturbations throughout the year of 2016. The deviations from average values are displayed. Surface emissivity variation at 23.8 GHz (lower left) shows potential melting events only during the summer months. The results are retrieved using our iterative algorithm. Skin temperature (lower middle) and precipitation of rain (lower right) from ERA5 are presented. The X-axis shows the month of the year, from 2016/1/1 to 2016/12/31.

### 4.4 More Comparison With Radiosonde Measurements Throughout the year

The Integrated Global Radiosonde Archive comprises radiosonde and pilot balloon observations from multiple stations distributed around the Antarctic coastline. Although the time resolution of these radiosonde measurements is lower than that of the ARM campaign, some stations provide data covering the entire year. Davis Station, located at 68.5744°S, 77.9672°E on the east coast, experiences significant variations in surface emissivity (Figure 13a). As shown in Figure 13c, the ATMS 3db beam at all emission angles (≤60°) partially intersects both the ground and sea ice. During the summer, the melting of sea ice causes a drop in emissivity. Our iterative algorithm can accurately determine the effective surface emissivity for observations where the beam covers mixed surface types, which otherwise for traditional retrieval process, the surface emissivity is unknown. This capability will benefit the atmospheric retrieval. Additionally, we detected two more melting events at SYOWA (Figure S.1) and MARIO ZUCHELLI Radiosonde Station (Figure S.2). When compared with *in situ* measurements from radiosondes, our iterative retrieval process efficiently captures all temperature and water vapor variations.



454 **DAVIS Radiosonde Station**

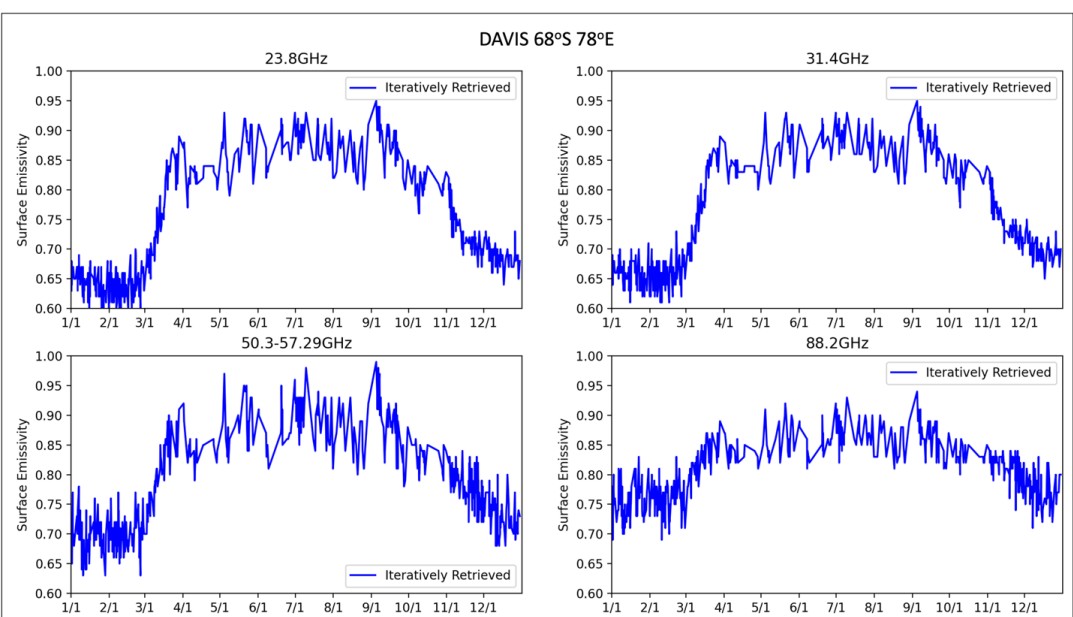

455

456 **Figure 13a.** Davis Radiosonde Station (68.5744°S, 77.9672°E). Surface emissivity from January 2016 to the end of

457 2016 at three window frequencies (23.8, 31.4 and 88.2 GHz) and the oxygen absorption band (50 – 58GHz). The blue

458 curve represents values from our iterative algorithm. The X-axis shows the month of the year, from 2016/1/1 to

459 2016/12/31.

460

461

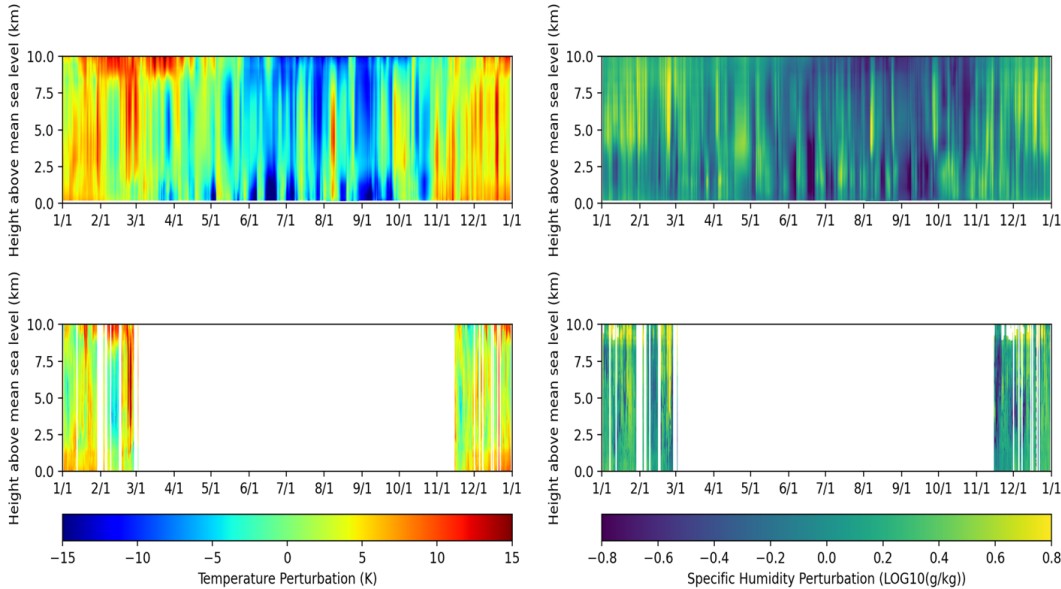

**Figure 13b.** Davis Radiosonde Station (68.5744°S, 77.9672°E). Temperature and specific humidity perturbations during 2016. The deviations from average values are displayed for that period. The top row presents results from our iterative algorithm, while the bottom row shows data from radiosonde measurements. The X-axis shows the month of the year, from 2016/1/1 to 2016/12/31.

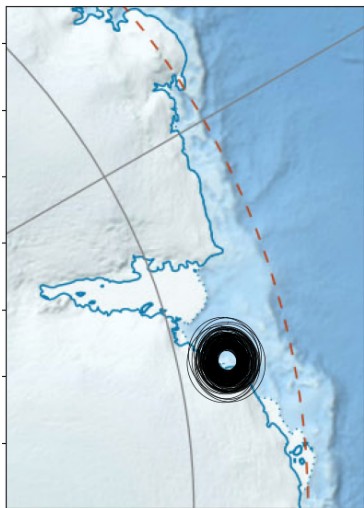

**Figure 13c.** ATMS 23.8GHz/31.4GHz 3db footprint at DAVIS radiosonde station at various emission angles ranging from nadir to 60°. The beam width is largest with 5.2° at 23.8GHz/31.4GHz. The beam width is 1.1° for channels in the 160-183 GHz range and 2.2° for the 80GHz and 50-60GHz channels. The elongation of the beam is proportional to $^1/_{cos^2\theta}$. At all emission angles



472

## 5    Conclusion

474

Atmospheric profile retrievals are highly sensitive to surface emissivity, necessitating a precision of approximately 0.02. Our iterative retrieval algorithm has demonstrated efficacy in accurately estimating temperature profiles, surface emissivity within 23 to 165 GHz, and detecting melting events. Nevertheless, the retrieval of water vapor content is intricately linked with surface emissivity at 183GHz. Thus, relying solely on this iterative atmospheric retrieval method may not yield precise water vapor content estimates. To address this issue, the atmospheric retrieval would need to be integrated with a surface emission estimate derived from an independent multi-frequency ice sheet melt, liquid water, temperature, and density retrieval algorithm. Independent estimation of ice sheet surface emissivity will help disentangle the impact of ice sheet emissions from atmospheric humidity.

The contact author has declared that none of the authors has any competing interests.

## Acknowledgements

This work is funded by the NASA Cryosphere Program. A contribution to this work was made at the Jet Propulsion Laboratory, California Institute of Technology, under a contract with National Aeronautics and Space Administration.

487

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
