# Peer review of "Retrieval of Atmospheric Water Vapor and Temperature"

_EGUsphere, 2024_

## Referee Comment (RC2)

Review for „**Retrieval of Atmospheric Water Vapor and Temperature Profiles over Antarctica through Iterative Approach**" submitted to *Atmospheric Measurement Techniques* by Zhang et al.

**General comments:**

The authors provide a study to determine atmospheric profiles of water vapor and temperature from satellite observations over Antarctica with the challenge of correctly assessing surface emissivity.

The study has a great potential in better understanding surface and atmospheric processes over the Antarctic continent by using microwave satellite observations. However, the current version of the manuscript still needs improvements, especially with respect to the description of the algorithms applied to satellite data as well as the use of ground truth, such as radiosondes and ground-based remote sensing observations.

Section 4 (Validation and Discussion) is very repetitive. Please consider only describing one or two regions in detail, whereas only mentioning the particular differences for the other regions. Some figures could be then moved to the annex.

Section 5 (Conclusions) is very short, it needs to clearly summarize the novelties of this study, also discussing the differences between the regions. Furthermore, it is necessary include a thorough discussion and quantification of accuracies, uncertainties and limitations.

**Specific comments:**

Line 30: Please clarify already here for "atmospheric retrievals" the variables that you are talking about. I guess it's about temperature and humidity, and maybe specify also that you are looking at the microwave range

Line 48: delete "a"

Lines 59 and 62: Do you mean Figure 1 here?

Lines 65-70: Beam width as well as ground-resolutions should be added to table 1

Line 82: You write: "Jacobians of the temperature and water vapor channels peak when the opacity becomes one." This is wrong: Jacobians peak at the maximum of the weighting functions.

Line 98: What is your motivation for Figure 2? Please start with a reasoning here

Figure 2: The plots are too small, please provide larger ones. Y-axis in subplots in 4th column should be limited to 20 km, as one cannot recognize much

Line 124: OE doesn't provide the most "efficient" solution, but the most "probable".

Lines 125-126: It is written "… particularly when retrieving temperature and humidity profiles from Microwave Radiometer (MWR) data". It does not matter which input you use, OE always provides the most probable solution.

Line 136: This section is not only about radiosonde data, you also use ground-based remote sensing data from ARM

Line 140: "albeit at a relatively low time resolution": I don't get what you want to say here. Do you mean that the radiosonde data are coarsely resolved vertically? Or that you have only one sonde per day?

Line 143: "AWARE collected atmospheric profiles every minute...": This are not radiosonde data, but ground-based remote sensing from ARM. It is necessary to describe this dataset in detail, especially also about its differences to radiosonde data (resolution, assumptions, etc.)

Line 148: By "altitude" resolution, do you mean "vertical" resolution?

Line 154: Do you use the radiosonde profile climatology as prior in the OE retrieval? If so, please mention this!

Table 2: How many radiosondes were used from which station? And for which time period? The sounding stations have different altitudes above sea level. Does this have an impact on your dataset?
The second part of table 2 concerning ARM data needs to be treated differently. Please mention which instruments were used here!

Line 174: Equation 1 is not necessary here. As you can't influence NEDT, just describe what components contribute to the noise.

Figure 3: Please refer to the variable names in the figure caption ($\varepsilon_0$, $\varepsilon_i$)

Figure 4: Which data were used to produce this figure? Over which time? Which area? All Antarctica? Is it from Ferraro et al., 2016? Or did you produce this figure?

Line 216: Either present full equation, or leave it out.

Lines 217-218: I don't understand this. What is $Q_{j\_Ref}$ ? Is it reference water vapor in a certain height? But if so, what is then $Q_{j\_H_2O}$?

Lines 241-248: This section about the radiative transfer equation should come earlier.

Line 252: "solve" instead of "derive"

Line 254 and Figure 5: Do you set the 6th iteration as reference? Otherwise, I don't know why the reference should be matching exactly the red line

Figure 5: "Retrieved using radiosonde derived emissivity" > this is not correct: you don't derive the emissivity from the radiosonde profiles. you just use the radiosonde profile as truth for the atmosphere

Line 281: What are the "optimal profiles"? Do you mean the profiles using surface emissivity from optimal estimation with radiosondes?

Line 287 (and later): "... with only 68% showing a difference within 30% of the actual profile." This is a very complicated way to say that "the standard deviation is 30%". Please simplify throughout the manuscript

Line 300: What is the "true surface emissivity from Radiosonde measurements"? This makes no sense to me.

**Technical comments:**

Please be consistent throughout the manuscript, how to name different products, such as retrieved profiles with OE, observed radiosonde profiles, etc.

The use of the language also needs to be improved, mostly in terms of using more precise expressions, but also a better use of English language in general.

Please add table captions

Check citation styles

---

## Author Response (AR1)

We thank the Editor and Reviewers for thoughtful and constructive comments, which have significantly improved the clarity and overall quality of the manuscript. Below, we provide a detailed response to each point raised. Changes in the manuscript have been carefully implemented and are indicated accordingly. We hope that the revised version addresses all concerns and meets the expectations of the reviewer. Thank you!
* * *
Editor

Upon reading your responses, I did not find any responses to RC1's general comments (about making the results more quantitative, discussing how clouds are included in the retrieval algorithm, and adding analysis for the low to mid troposphere).

Reply:

Thank you for your feedback. We would like to clarify that the points raised in RC1's general comments have been addressed in the manuscript and are detailed in our replies to specific reviewer comments. Below is a summary:

1. Quantitative Results

In response to the recommendation to make the results more quantitative, we have added a vertical profile of RMSE between satellite-derived and radiosonde temperature and humidity measurements at McMurdo. This addition, shown in Figure 7, addresses the request and illustrates retrieval performance in the low- to mid-troposphere.

2. Clouds in Retrieval Algorithm

The treatment of clouds in the retrieval has been incorporated and discussed in Section 2.2 of the revised manuscript. Specifically, we now explain that in addition to temperature and humidity, other MERRA-2 state variables—including 3D ice and liquid water cloud content—are assimilated into the retrieval process. This revision corresponds to RC1's comment under "Section 3: It is necessary to explain how clouds are included in the retrievals."

3. Low- to Mid-Troposphere Analysis

The added RMSE profiles (Figure 7) cover performance down to the near-surface, providing an assessment of the retrievals in the low- to mid-troposphere as requested.
* * *
Reviewer1

- *Abstract*: In the title and/or abstract there should be a mention that these are satellite retrievals.

Reply: We have made the following changes

1. Title changed to "Retrieval of Atmospheric Water Vapor and Temperature Profiles over Antarctica from Satellite Microwave Observations Using an Iterative Approach"
2. Abstract: Changed to "Retrieving atmospheric water vapor and temperature profiles over land using microwave radiometry is challenging due to uncertainties in estimating surface emissions."

- *Abstract, line 12*: Should the acronym ATMS on line 12 be spelled out?

Reply: Changed to "Using sounding channels from Ka- to G- band on the Advanced Technology Microwave Sounder (ATMS),....."

- *Introduction, line 42:* "With a total of 22 channels they are equipped..." The sentence needs revising.

Reply: Changed to "ATMS has 22 channels to receive and measure radiation from different layers of the atmosphere."

- *Section 2.1, line 55*: "...Sounder...are meant..." Should this be "is" meant?

Reply: Changed to "The Advanced Technology Microwave Sounder (ATMS), on board polar-orbiting satellites, is designed to perform atmospheric temperature and humidity sounding through oxygen and water vapor absorption bands."

- *Section 2.1, line 98*: For the Jacobians shown here what is the vertical grid used? Is it the same as the MERRA mentioned later (73 layers, between surface and 70 km)? What vertical resolution? Is it uniform grid or it varies with height.

Reply:

1. Changed to "The Jacobians are calculated by increasing the water vapor and temperature at each vertical layer to the orange dashed line, using the same 73-layer vertical grid as MERRA profiles (see Section 2.2), ranging from two meters above the surface to over 70 km in altitude."

2. Added "We use the "tavg3_3d_asm_Nv" dataset, which offers three-hourly averaged atmospheric profiles across 72 hybrid sigma-pressure levels [Rienecker et al. 2008],…"

We also added "The vertical grid is close to equidistant in the logarithm of the pressure and approximately equidistant in altitude."

- *Section 2.2.* I wonder if more discussion is needed here. What is used in the MERRA reanalysis product? It may be necessary to show at least the variability of the a priori profiles for a few locations. In my opinion in section 4 both MERRA profiles and these retrievals should be compared with radiosondes at McMurdo.
- *Section 3:* In this section it is necessary to explain how clouds are included in the retrievals

Reply: We have expanded Section 2.2 to include further discussion of the MERRA-2 product used as a priori information in the retrieval process. In response to your recommendation, we now include Figure 3, which presents the seasonal average a priori profiles of temperature and specific humidity for three Antarctic locations (McMurdo Station, South Pole, and Antarctic Peninsula), illustrating both spatial and temporal variability. This addition helps visualize the nature of the prior information and supports the importance of region- and season-specific context. We also briefly discuss the role of the a priori covariance matrix, which is also an important component of the Optimal Estimation framework and influences the retrieval characteristics.

In addition to temperature and humidity, other state variables derived from MERRA-2—including the 3D distributions of ice and liquid water cloud content, skin temperature, surface pressure, and wind speed—are also incorporated in the retrieval and are now discussed in Section 2.2.

- *Section 2.3, line 143*: "AWARE collected atmospheric profiles every minute". I think there is some confusion here and also regarding the WAIS deployment. During both deployments 4-6 radiosondes were launched daily. Here you are probably referring to the interpolated sonde product where the profiles between radiosonde launches are interpolated into a 1-minute time grid according to some scheme and then scaled using the ground-based microwave radiometer integrated water vapor.

Reply: We have revised the relevant paragraph in Section 2.3 to clarify that the 1-minute atmospheric profiles from AWARE are not direct radiosonde observations, but rather interpolated profiles derived from radiosonde launches and adjusted using integrated water vapor measurements from microwave radiometers. The revised text now makes a clear distinction between the radiosonde-based profiles (launched 4–6 times per day)

and the higher-frequency interpolated product, which is derived under assumptions of horizontal homogeneity and radiative transfer modeling.

- *Section 4:* Here or perhaps in the retrieval session it would be good to explain the temporal resolution of the satellite dataset. How many satellite retrievals per day do you get? What time of day? Does the time of day of the retrievals varies depending on location?

Reply: Added " The satellite carrying ATMS follows a sun-synchronous polar orbit, completing about 14 orbits per day and enabling frequent revisit opportunities, especially at high latitudes. Over the Antarctic region, this results in 4 to 10 ATMS overpasses per day for any fixed location, with variations due to swath width (~2,500 km), scan angle geometry, and Earth rotation." When introducing the observation data in Section2.1

- *Section 4.1, line 287:* "...only 68% showing a difference within 30% of the actual profile". This sentence is not clear. Why are the differences in specific humidity expressed as a percentage instead of g/kg (just like Kelvin for temperature?). It can be then specified that a given amount corresponds to 30%.

Reply: We chose to express differences in specific humidity as percentages rather than absolute values (e.g., g/kg) because specific humidity decreases by approximately two orders of magnitude within the first 10 km above the surface. Using absolute differences would disproportionately emphasize errors near the surface and underrepresent differences at higher altitudes. The percentage-based metric provides a more balanced view of profile-wide deviations. For clarity, we have revised the Figure.7 caption and added a note to explain this choice.

"the right panel shows specific humidity differences below 10 km, expressed as a percentage relative to the reference profiles. We use percentage differences for specific humidity to account for its strong vertical gradient, which spans approximately two orders of magnitude; this approach avoids biasing the evaluation toward near-surface values."

- *Fig. 8b and other similar figures:* It would be good to keep the same color palette (viridis?) for temperature and humidity. Again, it is not clear what is the temporal resolution of the retrievals. In the caption it should be specified if those are interpolated sondes (1-minute time resolution).

Reply: We chose to use different color palettes for temperature and humidity to help readers visually distinguish between the two variables more easily. Regarding temporal resolution, we have added clarification to the figure caption: "The temporal resolution in

the map is based on interpolation of ATMS observation time intervals, which are approximately a few hours apart."

- *Section 4.1, line 320:* "From the average value during the specified period". So, average between December 1 and March 1? Because this section and the next section refer to the same melt event it may be better to show both figures spanning the same time frame (Jan 1-30). This is also going to provide a comparable magnitude of the perturbation.

Reply: We appreciate the reviewer's suggestion for consistency in the time frame. We have now limited the time frame from 2016/01/01 to 2016/03/31 for consistency. We also clarified the period by adding "during the specified period (from 2016.01.01 to 2016.03.31)" to the figure caption.

- *Section 4.2 title*: This title is somewhat confusing. It says, "More locations near Ross ice shelf" but then there is a subsection titled "Ross ice shelf". Perhaps "WAIS divide" can be 4.2 and Ross ice shelf could be section 4.3. The same for the Ross shelf.

Reply: The revised Section 4.1, titled "Mid-January 2016 Melting Event Near the Ross Ice Shelf," now consolidates the analysis of McMurdo, WAIS Divide, and Ross Ice Shelf, all of which experienced varying degrees of surface melting and atmosphere profiles variation during that period.

We also added an introduction paragraph before discussion of each location:

"In this section, we examine the atmospheric and surface conditions during the mid-January 2016 melting event by analyzing retrieval results at three key Antarctic sites—McMurdo Station, WAIS Divide, and Ross Ice Shelf—over the same time period, each of which shows corresponding variations in atmospheric profiles, surface emissivity, or both during the melt."

Section 4.2, "West & East Coast Ice Shelves During 2016," focuses on the comparison of surface emissivity variability between the eastern and western Antarctic coasts. All other supporting results have been moved to the Supplementary Material for conciseness.

- *Fig. 9a:* It may be better to focus this on January 2016 as in figure 9b.

Reply: We have combined Figures 9a and 9b into a single Figure.10, now only focused on the time frame of January 1–30, 2016, for consistency.

- *Fig. 11 caption*: Is the average over the entire year? Because then you have the annual cycle embedded in the perturbations. See comment below

- *Figure 11 and 12 and 13* are somewhat difficult to see and their usefulness is not very clear. Perhaps a plot showing the retrieved emissivity vs skin temperature and retrieved emissivity vs. temperature/humidity perturbation may provide better insight. Also, I wonder if it is better to look at perturbations over summer/winter averages rather than all year average. Or you may want to low-pass filter (for example with a window of 2 months) the year time series to get the background field and subtract that from the single retrievals.

Reply: We have combined the emissivity data previously shown in Figures 11 and 12 into a single Figure.12a to better emphasize the contrast in emissivity variability between East and West Antarctic coastal ice shelves. For the West Coast shelves—where multiple potential melting events were identified—we now present atmospheric profile perturbations divided into three seasonal windows: January–March, May–June, and October–December. This seasonal breakdown allows clearer interpretation of variability without embedding the full annual cycle, as previously noted.
* * *
Reviewer2

- *Section 4 (Validation and Discussion) is very repetitive. Please consider only describing one or two regions in detail, whereas only mentioning the particular differences for the other regions. Some figures could be then moved to the annex.*

Reply: We have now focused the main text on two key aspects of the study:

1)  Section 4.1, titled "Mid-January 2016 Melting Event Near the Ross Ice Shelf," now consolidates the analysis of McMurdo, WAIS Divide, and Ross Ice Shelf, all of which experienced varying degrees of surface melting and atmosphere profiles variation during that period.

    We also added an introduction paragraph before discussion of each location:

    "In this section, we examine the atmospheric and surface conditions during the mid-January 2016 melting event by analyzing retrieval results at three key Antarctic sites— McMurdo Station, WAIS Divide, and Ross Ice Shelf—over the same time period, each of which shows corresponding variations in atmospheric profiles, surface emissivity, or both during the melt."

2)  Section 4.2, "West & East Coast Ice Shelves During 2016," focuses on the comparison of surface emissivity variability between the eastern and western Antarctic coasts.

All other supporting results have been moved to the Supplementary Material for conciseness.

- *Section 5 (Conclusions) is very short, it needs to clearly summarize the novelties of this study, also discussing the differences between the regions. Furthermore, it is necessary include a thorough discussion and quantification of accuracies, uncertainties and limitations.*

Reply: We have revised Section 5 (Conclusions) to clearly summarize the key novelties of our study. We have also expanded the discussion to include a thorough assessment and quantification of model accuracies, uncertainties, and limitations, as suggested.

- *Line 30: Please clarify already here for "atmospheric retrievals" the variables that you are talking about. I guess it's about temperature and humidity, and maybe specify also that you are looking at the microwave range*

Reply: We have made the changes: "Current atmospheric temperature and humidity retrievals in the microwave spectral range face challenges…"

- *Lines 65-70: Beam width as well as ground-resolutions should be added to table 1*

Reply: We have updated Table 1 to include both the beam width and nadir ground resolution for each channel, providing a more complete summary of the instrument's spatial characteristics.

- *Line 82: You write: "Jacobians of the temperature and water vapor channels peak when the opacity becomes one." This is wrong: Jacobians peak at the maximum of the weighting functions.*

Reply: We agree with the reviewer that our original wording was imprecise. We have revised the sentence to correctly reflect the physics:

"Jacobians of temperature and water vapor channels typically peak near the maximum of the weighting function, which often corresponds approximately to the atmospheric level where the optical depth is near unity."

- *Line 98: What is your motivation for Figure 2? Please start with a reasoning here*

Reply: We have now started the paragraph with "To better understand the vertical sensitivity of each ATMS channel to atmospheric temperature and water vapor, we examine the Jacobians, which describe how changes in these variables influence observed brightness temperature."

- *Figure 2: The plots are too small, please provide larger ones. Y-axis in subplots in 4th column should be limited to 20 km, as one cannot recognize much*

Reply: In the revised manuscript, we have updated Figure 2 to only display results from a representative summer (December) atmosphere profile. We have enlarged all subplots for improved readability, increased label font sizes, and adjusted the y-axis range in the humidity-related plots to a maximum of 20 km.

- *Line 124: OE doesn't provide the most "efficient" solution, but the most "probable".*

Reply: We have revised the text as suggested.

- *Lines 125-126: It is written "... particularly when retrieving temperature and humidity profiles from Microwave Radiometer (MWR) data". It does not matter which input you use, OE always provides the most probable solution.*

Reply: We have revised the sentence to:

"Based on Bayes' theorem, OE estimates the most probable atmospheric state."

- *Line 136: This section is not only about radiosonde data, you also use ground-based remote sensing data from ARM*

Reply: We have changed the section title to "2.3  In Situ and Ground-Based Remote Sensing Observations"

- *Line 140: "albeit at a relatively low time resolution": I don't get what you want to say here. Do you mean that the radiosonde data are coarsely resolved vertically? Or that you have only one sonde per day?*

Reply: We agree that the original sentence was unclear. We have revised it to specify that the limitation refers to low temporal resolution. And this section has been moved to the supporting materials.

"These stations contribute to the Integrated Global Radiosonde Archive (IGRA), which provides valuable long-term in situ measurements, though often at limited temporal resolution depending on station operations"

- *Line 148: By "altitude" resolution, do you mean "vertical" resolution?*

Reply: Yes, we were referring to vertical resolution, and have updated the manuscript accordingly by replacing "altitude resolution" with "vertical resolution" to ensure consistent and accurate terminology.

- *Line 154: Do you use the radiosonde profile climatology as prior in the OE retrieval? If so, please mention this!*

Reply: We now clarify that we did not use radiosonde profile climatology as the prior in the OE retrieval. The iterative retrieval algorithm was developed to operate across the entire Antarctic region, including areas where radiosonde data are unavailable. Since radiosonde observations are only available at a limited number of coastal stations, they are used solely for validation purposes in our study. We have updated the manuscript accordingly to explicitly state this point.

"Importantly, radiosonde profiles were used solely for validation and not as climatological priors in the retrieval process, as our goal is to enable retrievals across all of Antarctica, including areas without radiosonde coverage."

- *Table 2: How many radiosondes were used from which station? And for which time period? The sounding stations have different altitudes above sea level. Does this have an impact on your dataset? The second part of table 2 concerning ARM data needs to be treated differently. Please mention which instruments were used here!*

  *Line 143: "AWARE collected atmospheric profiles every minute...": This are not radiosonde data, but ground-based remote sensing from ARM. It is necessary to describe this dataset in detail, especially also about its differences to radiosonde data (resolution, assumptions, etc.)*

Reply: In our study, we evaluated the performance of the iterative retrieval algorithm using data from 5 radiosonde stations across Antarctica throughout the year 2016. The exact temporal coverage from each station is now detailed in Table 2. We have also clarified in the revised table that the second section concerns data obtained from the ARM West Antarctic

Radiation Experiment (AWARE) campaign, which relies on ground-based remote sensing instruments, not direct radiosonde measurements.

We have now revised Section 2.3 of the manuscript to describe the AWARE dataset in more detail. Specifically, we clarify that the high-temporal-resolution atmospheric profiles were generated using interpolated radiosonde launches combined with ground-based microwave radiometer observations, and scaled based on integrated water vapor retrievals. We also discuss the assumptions made in generating these profiles and contrast their vertical and temporal resolutions with those of traditional radiosonde observations.

Regarding station altitude, we acknowledge that different sounding stations are located at various elevations above sea level, leading to differences in surface pressure, which can impact radiative transfer calculations. In our retrieval framework, surface pressure and altitude from MERRA2 reanalysis is used as an input to the forward model to account for these variations. This ensures that the influence of altitude on the radiative signals is appropriately represented in both the simulation and the retrieval process.

- *Line 174: Equation 1 is not necessary here. As you can't influence NEDT, just describe what components contribute to the noise.*

Reply: We have removed Equation 1 from the manuscript and revised the corresponding text to focus on a qualitative description of the components that contribute to NEΔT. Specifically, we now added

"The noise equivalent delta temperature (NEΔT) represents the level of instrument noise inherent in brightness temperature measurements. For a total power radiometer like ATMS, NEΔT is influenced by several system-level factors, including the system noise temperature (which incorporates atmospheric and instrumental contributions), the radiometer's bandwidth, the integration time, and fluctuations in instrument gain."

- *Figure 4: Which data were used to produce this figure? Over which time? Which area? All Antarctica? Is it from Ferraro et al., 2016? Or did you produce this figure?*

Reply: We have modified the caption and main text to specify that Figure 4 (now Figure 5) was produced using archived AMSU-A emissivity data at 23, 31, and 50 GHz, covering the entire Antarctic region during the year 2016. These emissivity values were originally computed by Ferraro et al. (2018) and distributed as part of the AMSU-A data archive. The

figure itself was created by ourselves, not directly taken from their publication. And we now reference Ferraro et al. (2018).

- *Line 216: Either present full equation, or leave it out.*

Reply: We have removed the equations.

- *Lines 241-248: This section about the radiative transfer equation should come earlier.*

Reply: The radiative transfer equation has now been moved to Section 3.2 Forward Model.

- *Line 252: "solve" instead of "derive"*

Reply: We have changed accordingly.

- *Line 254 and Figure 5: Do you set the 6th iteration as reference? Otherwise, I don't know why the reference should be matching exactly the red line*

Reply: No, we did not set the 6th iteration as a reference. The example shown was chosen because it demonstrates a particularly good fit — in this case, the iteratively retrieved surface emissivity aligns closely with values derived from ARM observations. We've clarified this in the revised caption of Figure 6 to indicate that this is a representative case of a well-fitting result, and not all iterations achieve the same level of agreement.

Figure5 Caption: "This demonstrated example is a particularly good fit where the retrieved surface emissivity closely matches ARM-derived values; not all cases show this level of agreement."

We also acknowledge that our approach does not always yield equally accurate results: "However, at 183 GHz - the water vapor absorption band - surface emission is tightly coupled with near-surface humidity, making accurate emissivity retrieval more difficult. As a result, the iterative method may yield less reliable results at this frequency (see Section 4.1). "

- *Figure 5: "Retrieved using radiosonde derived emissivity" > this is not correct: you don't derive the emissivity from the radiosonde profiles. you just use the radiosonde profile as truth for the atmosphere*
- *Please be consistent throughout the manuscript, how to name different products, such as retrieved profiles with OE, observed radiosonde profiles, etc.*
- *Line 281: What are the "optimal profiles"? Do you mean the profiles using surface emissivity from optimal estimation with radiosondes?*

Reply: We have revised the manuscript to ensure consistency in terminology across all relevant sections and figures.

To maintain clarity, we now use the following standardized terms throughout the manuscript:

1) Truth profiles: In-situ and ground-based measured atmospheric profiles

Truth profiles have high vertical resolution

2) Reference surface emissivity: Surface emissivity calculated using in-situ and ground-based measured atmospheric profiles.

Reference atmospheric profiles: Atmospheric profiles retrieved using the Optimal Estimation (OE) method, assuming the reference surface emissivity.

Reference atmospheric profiles represent the best achievable atmospheric retrievals from the 22-channel ATMS observations and serve as a benchmark for evaluating our iterative retrieval performance. Reference atmospheric profiles have lower vertical resolution as compared to the Truth profiles.

3) Iteratively retrieved products: Atmospheric profiles and surface emissivity retrieved jointly using OE, without prior knowledge of the surface emissivity, via an iterative approach.

- *Line 287 (and later): "... with only 68% showing a difference within 30% of the actual profile." This is a very complicated way to say that "the standard deviation is 30%". Please simplify throughout the manuscript*

Reply: We have simplified the language throughout the manuscript as suggested. All related phrases have been revised to clearly state the standard deviation values instead of using percentile-based descriptions.

- *The use of the language also needs to be improved, mostly in terms of using more precise expressions, but also a better use of English language in general.*

Reply: We have reworded several sections of the manuscript to improve clarity, precision, and overall language quality. Particular attention was given to refining technical expressions, correcting grammatical issues, and ensuring consistency in scientific terminology throughout the text.

- *Please add table captions*

Reply: We have added captions for all tables.

Thank you!